# Techno-Economic Assessment of Bio-Energy with Carbon Capture and Storage Systems in a Typical Sugarcane Mill in Brazil †

**Sara Restrepo-Valencia * and Arnaldo Walter**

Department of Energy, School of Mechanical Engineering, University of Campinas—UNICAMP, Campinas 13083860, Brazil; awalter@fem.unicamp.br
* Correspondence: sara.valencia@fem.unicamp.br; Tel.: +55-19-352-13283
† The present work is an extension of the paper "A. Techno-Economic Assessment of BECCS Systems in the Brazilian Sugarcane Sector" presented at the 13th Conference on Sustainable Development of Energy, Water and Environment Systems—SDEWES Conference, 30 September–4 October, Palermo, Italy.

**Abstract:** For significantly reducing greenhouse gas emissions, those from electricity generation should be negative by the end of the century. In this sense, bio-energy with carbon capture and storage (BECCS) technology in sugarcane mills could be crucial. This paper presents a technical and economic assessment of BECCS systems in a typical Brazilian sugarcane mill, considering the adoption of advanced—although commercial—steam cogeneration systems. The technical results are based on computational simulations, considering $CO_2$ capture both from fermentation (released during ethanol production) and due to biomass combustion. The post combustion capture technology based on amine was considered integrated to the mill and to the cogeneration system. A range of energy requirements and costs were taken from the literature, and different milling capacities and capturing rates were considered. Results show that $CO_2$ capture from both flows is technically feasible. Capturing $CO_2$ from fermentation is the alternative that should be prioritized as energy requirements for capturing from combustion are meaningful, with high impacts on surplus electricity. In the reference case, the cost of avoided $CO_2$ emissions was estimated at 62 €/t $CO_2$, and this can be reduced to 59 €/t $CO_2$ in case of more efficient technologies, or even to 48 €/t $CO_2$ in case of larger plants.

**Keywords:** bioelectricity; carbon capture; negative emissions; sugarcane; biomass; climate change

## 1. Introduction

In order to maintain 2 °C as the maximum increase in the global average temperature, the levels of atmospheric concentrations must be kept below 450 ppm of $CO_{2eq}$ during the 21st century [1]. Therefore, worldwide emissions of $CO_2$ have to be drastically reduced in the coming decades, inducing deep changes in the energy systems [2]. This scenario requires that emissions from electricity generation should be negative by the end of the century, with fast progress in energy efficiency and promotion of low-carbon technologies. In this context, carbon capture and storage (CCS) is crucial because it represents a process by which large amounts of carbon dioxide can be captured and stored for the long term [1].

The CCS technology involves four main steps: conditioning processes to separate $CO_2$ into a pure stream, carbon capture itself, its compression and, finally, storage for long term periods [1]. In the case of CCS applied to power units, significant losses in efficiency are expected; for instance, the Intergovernmental Panel on Climate Change (IPCC) indicates a 9% net reduction in efficiency for coal-fired power plants (pulverized) and 7% for combined cycle gas-fired power plants [1].

Post-combustion technology consists in the removal of $CO_2$ from the exhaust gases. Capture by absorption is recognized as the reference technology [3,4] and considered mature for power plants [5]. The removal from flue gases uses a solvent, generally amines, to absorb $CO_2$ molecules, being $CO_2$ then released by heating or drastic pressure reductions [6]. Flue gases need to be cooled before getting in contact with the solvent: the temperatures must be between 40 and 60 °C at the entrance of absorption columns [7]. Costs and energy requirements—also called energy penalties—from a CCS unit using absorption capture depends mainly on the solvent properties. It is estimated that the heating for solvent regeneration is responsible for over 25% of the energy penalty when compression is included [7].

Combining bio-energy with carbon capture and storage (BECCS) offers the prospect of energy supply with net negative emissions and is clearly an important approach to reach the target of 2 °C. BECCS combines production of fuels and electricity from renewable biomass with carbon capture and storage of the $CO_2$ emitted when biomass is converted [2]. As the $CO_2$ is removed from the atmosphere during the growth of the raw material, life-cycle absolute emissions of BECSS could be negative [8]. In this sense, BECCS technology applied in sugarcane mills would be fundamental, contributing with very low greenhouse gas (GHG) emissions in both the transport sector (with avoided emissions due to the displacement of fossil gasoline) and in electricity generation [9].

The production of ethanol (via the fermentation of sugars) releases a pure stream of $CO_2$, which means there is no penalty for its separation in the CCS process. This is the most obvious option to capture $CO_2$ in sugarcane mills, and it is estimated that, considering the ethanol production figures of 28.5 million $m^3$ in Brazil, it would be possible to reduce $CO_2$ emissions by 27.7 million tonnes per year [9]. Carbon capture in sugarcane mills could at least double—or triple—with the adoption of CCS technologies in cogeneration systems in which residual biomass is burned—usually bagasse, and more recently, bagasse combined with straw.

This work focuses on assessing the technical and economic impacts of BECCS in a typical Brazilian sugarcane mill. First, a hypothetical sugarcane mill was selected to perform the evaluation that considers advanced steam cogeneration systems. A literature review was conducted to select the CCS technology and obtain representative data to model the integration of the CCS unit with the cogeneration plant. The feasibility analysis is based on typical costs (i.e., investments, operation and maintenance costs) and efficiencies, and the final assessment is based on the costs of $CO_2$ avoided emissions.

## 2. Materials and Methods

### 2.1. Cogeneration Plant

A typical Brazilian sugarcane mill, but rather representative among the more than three hundred existing mills, was considered for this study, with a 4 Mt/y (million tonnes of sugarcane crushed per year) milling capacity. The cogeneration unit would be fully integrated to the mill to supply electric power and steam to the industrial process, and also maximizing surplus electricity. The steam demand for both sugar and ethanol production was assumed equivalent to 340 kg of steam (at 2.5 bar and 137 °C) per tonne of sugarcane; this is the minimum consumption of steam currently considered economically viable [10]. The power plant would operate along the whole year (with 90% capacity factor), being as a cogeneration unit during the harvest season and as a single power plant during off-season. Biomass would be stored to assure the operation during off-season and this is already a common practice for mills that generate electricity throughout the year; in general, mills have area available for this, and the costs are not prohibitive. The cogeneration technology is the one known as condensing-extraction steam-turbine (CEST), with live steam at the highest possible pressure and temperature (120 bar/535 °C is the state-of-art in Brazilian sugarcane mills, according to [9]).

The CEST technology is very common in modern sugarcane mills. Bagasse used to be the only fuel but, recently, a blend of bagasse and straw has been used [11] due to the growing straw availability at the mill site as a consequence of mechanized harvesting. Table 1 presents a summary of the main characteristics

of the reference mill operating with CEST system and burning biomass—bagasse and straw—as fuel. Bagasse availability is defined by the fiber content of the sugarcane plant (14%), i.e., 280 kg of bagasse with 50% moisture per tonne of cane. As for straw its availability at the mill was considered 50% in relation to the total amount available at the field, resulting 161 kg per tonne of cane (with 13% moisture).

**Table 1.** Characteristics of the reference mill and the power plant.

| Parameter | Value |
|---|---|
| Power plant annual capacity factor | 90% |
| Milling capacity (t/h) | 772 |
| Annual harvest season (h) | 5184 |
| Mill capacity factor during harvest season | 90% |
| Total annual milling capacity (Mt/y) | 4.0 |
| Bagasse availability per tonne of sugarcane [a] (kg) | 280 (50% moisture content) |
| Straw availability per tonne of sugarcane [b] (kg) | 161 (13% moisture content) |
| Energy demand | |
| Steam process requirement per tonne of sugarcane (kg) | 340 |
| Electricity consumption per tonne of sugarcane [c] (kWh) | 30 |
| Cogeneration system—CEST | |
| Boiler efficiency (base LHV) | 85% |
| Live steam parameters | 120 bar/535 °C |
| Isentropic efficiency of the steam turbine (per body) | 79% |

Sources: [a] [11] for bagasse's LHV 7.52 MJ/kg; [b] [11] for straw's LHV 12.96 MJ/kg; [c] [9].

The CEST system was modelled in a non-commercial software able to simulate its integration with sugarcane mills and to estimate electricity generation [12]. The current experience with straw use as fuel has shown that problems like slagging, fouling and surface corrosion are common when the straw share is above 15–20% in the fuel blend (mass basis). These problems are due to biomass and their ash compositions, which have much more chlorine, CaO and $K_2O$ in the case of straw compared to bagasse [13]. As it is predicted that in the considered case straw would represent about one third of the fuel input, the hypothesis is that the problems mentioned would be solved in the future. Biomass consumption—bagasse and straw—would be distributed along the year to assure fuel supply according to system's requirement.

The emissions of $CO_2$ from combustion were estimated considering full combustion with 30% excess air [14], and carbon content 48.6% in the dry fuel for both bagasse and straw [15]. For estimating $CO_2$ from fermentation, it was assumed a sugar mill with a medium to large annexed distillery (i.e., 50% of the sugarcane would be used to ethanol production). A typical Brazilian mill with such a capacity (4 Mt/y) produces both ethanol and sugar with some flexibility (in general, each output varies between 40% and 60%; basis is the sugarcane input) [16]. Therefore, $CO_2$ from fermentation was calculated for an ethanol production of 86.3 L per tonne of sugarcane [17] and a $CO_2$ production of 0.96 kg per kg of ethanol [9], resulting in an emission index 0.78 kg of $CO_2$ per liter of ethanol (calculated for an ethanol density of 0.809 kg/L).

### 2.2. CCS Unit

Carbon dioxide both from fermentation and from biomass combustion were considered to be processed in a CCS unit. $CO_2$ from the combustion passes through the complete CCS process (referred to as absorption, regeneration and compression), while $CO_2$ from fermentation only passes through the compression steps. These two streams are combined in the transportation and storage stages. It was assumed a post-combustion technology based on capture with monoethanolamine (MEA).

### 2.2.1. MEA Technology

Capture process by absorption is the reference technology and MEA is the incumbent solvent used. Absorption characteristics of the solvent determine energy penalties and impact the economic feasibility

of capturing. In this study, parameters of the capture process based on MEA technology were taken from [7]. Solvent regeneration was considered using steam extracted at the medium-pressure stage of the steam turbine, that coincidentally is the same pressure required by the industrial process (i.e., 2.5 bar, 137 °C). The three levels of heat requirement for solvent regeneration are related with the different stages of technology development: 4.4 GJ/t $CO_2$ (1998 kg of steam per tonne of $CO_2$); 2.6 GJ/t $CO_2$ (1180 kg of steam per tonne of $CO_2$); and 1.6 GJ/t $CO_2$ (726 kg of steam per tonne of $CO_2$). It was assumed 90% as the maximum possible capture rate in the CCS unit. Absorption and regeneration take place in the unit hereafter referred to as CCS. Power requirement for flow gases treatment (at the CCS unit) was estimated at 25.84 kW per unit of exhaust gas flow, in kg/s [18]. This figure includes electricity requirement for pumps and blowers, capture pre-treatment pumping, cooling water pumping and blower duties and, finally, solvent pumping duties.

### 2.2.2. Compression Unit

After $CO_2$ separation from the exhaust gases, it goes to the compression unit, being its power requirement estimated from [19]. $CO_2$ is compressed from 1 bar to 150 bar in order to be transported through a pipeline. Compression is divided into two steps: first compression from 1 bar to the $CO_2$ critical pressure (73.9 bar), and then, in the liquid phase, a pump can be used to boost final pressure. First step was assumed as an ideal gas compression in 5-stages with intermediate cooling, and 85% isentropic efficiency per stage. Pumping requirement was calculated with isentropic efficiency of 85%. Power requirements were considered for each $CO_2$ stream.

### *2.3. Economic Performance Assessment*

Cost estimations were done based on a literature review for each technology: CEST system, CCS based on MEA technology, $CO_2$ compression and $CO_2$ transport and storage at a nearby saline aquifer. All costs are presented in €$_{2014}$ in order to be coherent with the references used for CCS systems. For all equipment the useful life is 25 years, and the discount rate considered in the base case is 10% per year. This discount rate was chosen because it is a compromise taking into account the investments on generating electricity with biomass—annual rates of less than 10% would make investments more difficult—and the investments on carbon capturing and storage—in this case, due to the technology stage, the feasibility is related to a discount rate as low as possible. In any case, the results for the discount rate of 8% per year are also presented (see Section 3.2) in order to make comparisons possible with what has been published. For electricity generation, the feasibility was evaluated based on the minimum selling price (MSP). In the case of $CO_2$ capture, the minimum credit price (i.e., the income obtained by selling the credits of capturing $CO_2$) was estimated to cover all costs, including electricity that is not sold due to energy sanctions imposed by CCS.

### 2.3.1. Power Plant Capital Costs

The cogeneration plant at the sugarcane mill aims at self-sufficiency and the sale of surplus electricity to the grid. Capital costs were estimated ($/kW) from an updated function adapted from [20], which estimates turn-key investments in Brazilian currency (R$), including storage of biomass and connecting costs to the grid, according to Equation (1). Values in R$$_{2014}$ were converted into Euro using the exchange rate by the end of 2014 (3.23 R$/€):

$$C_{CEST} = 3578 \cdot (capacity)^{-0.334} \tag{1}$$

where C represents the specific capital costs, in €/kW installed for the CEST technology, and capacity is the total installed *capacity* in MW.

### 2.3.2. CCS Unit and Compression Unit Capital Costs

For the CCS unit and the compression unit, scaling—according to Equation (2)—was used to estimate capital costs (scale factor 0.6); scaling is function of $CO_2$ capturing capacity ($CO_2$ flow going to CCS and to the compression unit). Values from [4] were taken to estimate the parameters of units based on MEA technology. The costs of the three different technology levels were considered:

$$C = C_{ref} \cdot \left( \frac{Q}{Q_{ref}} \right)^{\alpha} \tag{2}$$

where C represents the capital cost, Q the capacity, $\alpha$ is the scaling factor and ref indicates the reference case.

### 2.3.3. Transport and Storage Capital Costs

In this study, $CO_2$ storage was considered to be at the geological formation Rio Bonito, located in the south and southeast regions of Brazil [21]. $CO_2$ injection shall be at least 1200 m below surface [9]. It is assumed that geological conditions are adequate to keep $CO_2$ stored for centuries, as required to make CCS a real alternative to mitigate GHG emissions.

Capital costs for transport and storage were estimated from [9]. It was assumed a pipeline with 10 km length, hypothesis that is coherent with the assumption that sugarcane mills are located nearby existing saline aquifers. Storage capital costs include a preliminary assessment of three wells drilled at 1200 m deep, which is a practical assumption to find a reservoir with appropriate conditions to long term storage.

### 2.3.4. Fuel Costs

As it was mentioned in Section 2.1, bagasse and straw are the biomasses burned in the boiler. No cost was attributed to the bagasse, as it is already available at the mills. This assumption was taken to simplify the economic analysis: in specific cases mills have the opportunity to sell some surplus bagasse to other consumers, depending on the location and the amount available. For the use of straw, the combined cost of collecting and transport to the mill was attributed, summing-up 17.76 € per tonne of straw [22].

### 2.3.5. Operation and Maintenance Costs

Operation and maintenance costs (O&M) of $CO_2$ capture (CCS and compression units) are based on [4] and were estimated, as annual values, as function of the total investment. Annual O&M costs for the cogeneration plant were assumed according to the current practices in Brazil. For $CO_2$ transport and storage, as a simplification, annual O&M costs were assumed at 2% of the total investment. Table 2 presents assumptions for O&M in this study.

**Table 2.** Assumptions for operation and maintenance costs.

| Parameter | Annual Value as Function of the Total Investment |
|---|---|
| Cogeneration system—CEST | 2% |
| CCS unit | 5.8% |
| Compression unit | 4.6% |
| Transport and storage | 2% |

### 2.4. Scaling Effects

The previous sections presented the hypothesis for assessing the feasibility of BECCS in sugarcane mills. Scaling effects on milling capacity were explored as a significant impact on capital costs is expected. For this reason, it was also considered a smaller milling capacity (2 Mt/y) and a larger mill (8 Mt/y). Annual milling capacity of 2 Mt of sugarcane crushed could be considered as an average mill

in Brazil; [16] reports that 39% of sugarcane mills are close to this capacity. Bigger capacities, as 4 Mt/y and 8 Mt/y, are less usual—4 Mt/y is more common and few units are close to 8 Mt/y—but larger mills is the general tendency in the future.

*2.5. GHG Emissions Due to the Supply of Biomass*

It was assumed, by simplification, that both bagasse and straw are carbon neutral, i.e., there would be no GHG emissions due to the biomass used as fuel in the cogeneration unit. Many life cycle assessments of ethanol from sugarcane and electricity generation from bagasse assume that the bagasse is carbon neutral (it would be a residue) being all the environmental burdens imposed on ethanol and sugar, as the main final products [10,17,23]. In the case of straw, as currently there is no other use other than as fuel on the site of the mill, and because the straw is derived from a mechanized harvest that is a new practice, it is common to impose to the straw a share of the emissions of the sugarcane harvest and its transport to the mill. In this sense, the hypothesis assumed in this document is optimistic regarding the benefits of carbon capture related to cogeneration systems. However, it is important to bear in mind that approximately one third of the amount of straw that is supposed to be available as fuel at the factory site is anyway transported to the plant as impurities and, in addition, the straw represents approximately one third of the total energy input. Therefore, the simplification carried out does not imply a great distortion with respect to the benefits of carbon capture, and was considered reasonable for a preliminary evaluation of BECCS in a sugarcane mill.

## 3. Results and Discussion

This section is devoted to present and discuss the results and is divided into three parts. The first part presents the technical performance of the integrated BECCS systems to a hypothetical sugarcane mill. As previously mentioned, three levels of heat demand for solvent regeneration—related to the different stages of the technology—were evaluated. The second part focuses on the feasibility of carbon capture in a sugarcane mill. Finally, in the third part the effects of scale are analyzed.

*3.1. Technical Performance*

The simulated cogeneration system has a steam turbine with one controlled extraction at 2.5 bar and condensation of the remaining flow. Five cases were assessed: the reference case, i.e., the cogeneration plant without CCS; case 1—cogeneration plant with CCS only from $CO_2$ of fermentation; case 2—cogeneration plant with CCS from both fermentation and combustion, and solvent regeneration requiring 4.4 GJ/t $CO_2$; case 3—cogeneration plant with CCS (fermentation and combustion $CO_2$) and 2.6 GJ/t $CO_2$ as heat requirement; and case 4—cogeneration plant with CCS (fermentation and combustion $CO_2$) and 1.6 GJ/t $CO_2$ as heat requirement. For all cases, simulation includes harvest and off-harvest seasons.

Figure 1 shows the process flow diagram that represents the operation in the harvest season (i.e., with steam extraction for industrial process), with CCS. In software's basic configuration the steam turbine has three bodies. The steam flow to the deaerator (stream 1) corresponds to 2% of the steam raised and is extracted from the turbine body *b*. The stream (2) feeds the industrial process (stream 5) and the heat exchanger for regenerating the solvent (stream 4), being its thermodynamic state adjusted (in a desuperheater) to the required temperature (137 °C). Streams (6), (7) and (8) refer to condensates, being assumed 90% recovery of streams (6) and (8), but both at 90 °C; pumps for setting the pressure of condensing flows before the deaerator are omitted in Figure 1.

Performance results are presented in Table 3. In the reference case—cogeneration without $CO_2$ capture, there is no steam extraction going to the CCS plant and, therefore, power output is maximum. In this case, net power output was estimated at 77 MW during harvest season and 64 MW in the off-season, result that corresponds to a surplus output of 144 kWh per tonne of sugarcane (174 kWh/t generated). Electricity generation, or sold, per tonne of sugarcane crushed is an indicator commonly used to express the efficiency of electricity production in a sugarcane mill, and the result

above can be compared to the predicted current best figures in sugarcane sector (130–170 kWh/t of cane) [9,23].

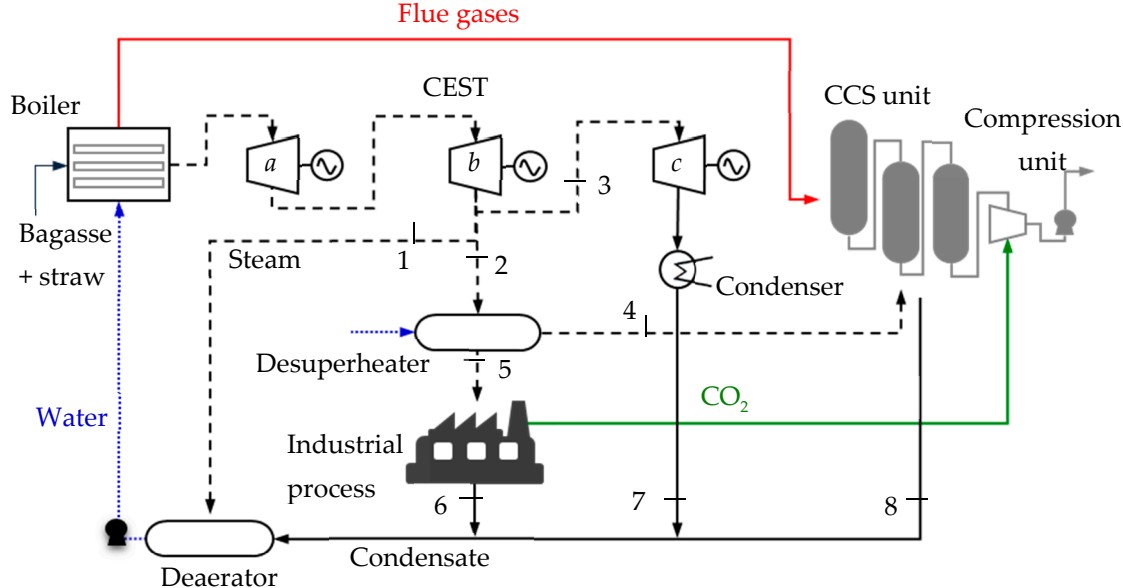

**Figure 1.** BECCS process flow diagram (harvest season).

**Table 3.** Performance results for the BECCS systems.

| Parameter | Reference Case | Case 1 | Case 2 | Case 3 | Case 4 |
|---|---|---|---|---|---|
| Energy (as steam) for regeneration (GJ/t $CO_2$) | - | - | 4.4 | 2.6 | 1.6 |
| $CO_2$ emission (Mt $CO_2$/y) | 1.38 | 1.25 | 0.58 | 0.29 | 0.12 |
| Total $CO_2$ captured (Mt $CO_2$/y) | - | 0.13 (10%) | 0.79 (58%) | 1.09 (79%) | 1.26 (91%) |
| Harvest season | | | | | |
| $CO_2$ captured (combustion) (Mt $CO_2$/y) | - | - | 0.43 (43%) | 0.72 (73%) | 0.88 (90%) |
| $CO_2$ captured (fermentation) (Mt $CO_2$/y) | - | 0.13 (100%) | 0.13 (100%) | 0.13 (100%) | 0.13 (100%) |
| Net power output (MW) | 100.6 | 100.6 | 85.3 | 85.3 | 88.8 |
| Mill demand (MW) | 23.2 | 23.2 | 23.2 | 23.2 | 23.2 |
| Power requirement for CCS unit (MW) | - | - | 11.4 | 19.3 | 23.6 |
| Compression power(combustion) (MW) | - | - | 7.6 | 12.9 | 15.8 |
| Compression power (fermentation) (MW) | - | 2.4 | 2.4 | 2.4 | 2.4 |
| Net power output (MW) | 77.4 | 75 | 40.7 | 27.5 | 23.8 |
| Off-season | | | | | |
| $CO_2$ captured (combustion) (Mt $CO_2$/y) | - | - | 0.24 (90%) | 0.24 (90%) | 0.24 (90%) |
| Power output (MW) | 64.1 | 64.1 | 48.8 | 55.0 | 58.6 |
| Power requirement for CCS unit (MW) | - | - | 12.3 | 12.3 | 12.3 |
| Compression power (MW) | - | - | 8.2 | 8.2 | 8.2 |
| Net power output (MW) | 64.1 | 64.1 | 28.3 | 34.5 | 38.1 |
| General results | | | | | |
| Total electricity output (GWh/y) | 695 | 682 | 408 | 356 | 346 |
| Surplus electricity output (GWh/y) | 575 | 562 | 288 | 236 | 226 |
| Surplus electricity per tonne (kWh/t) | 144 | 141 | 72 | 59 | 57 |
| Energy penalty due CCS | - | 2% | 43% | 50% | 52% |

Case 1 presents a special case in which capture of only $CO_2$ from fermentation was considered. As the $CO_2$ from fermentation is naturally a pure stream, and separation is not necessary, $CO_2$ goes directly to the compression unit. Power requirement for compression was estimated at 2.4 MW (2% of the net power output) and capture corresponds to 0.13 Mt $CO_2$/y, which means 10% of the total mill's emissions.

Capturing $CO_2$ from both fermentation and combustion imposes meaningful energy penalties. There is a constraint in Case 2—with 1998 kg of steam required per tonne of $CO_2$—due to steam availability: during harvest season the system is unable to supply all required steam for both the industrial process and solvent regeneration, thus forcing the reduction of the capture rate. In all

cases a minimum steam flow of 3 kg/s (stream (3)) was assumed to be expanded at *c*. Results of Case 2 indicate that it is possible to capture only 43% of the $CO_2$ emitted by the combustion process. However, during the off-season, as no steam is required for the industrial process, 90% of the $CO_2$ from combustion gases is captured (the maximum assumed). Thus, power required for compression during harvest season is due to all $CO_2$ from fermentation and to the amount captured from flue gases, while during off-season all possible $CO_2$ captured from flue gases (90%) is compressed. In summary, annual capture rate was estimated at 58%, capturing 0.79 Mt $CO_2$ (0.13 from fermentation), which is a significant value for a BECCS system, taking into account what has been considered. However, in this case it is predicted a significant reduction in net power output: 48% of the total power during harvest season and 59% otherwise. The balance corresponds to the surplus of 40.7 MW and 28.3 MW, in the harvest and the off-season, respectively. Even though, results still correspond to meaningful surplus electricity regarding the current practices: 72 kWh/t of cane.

In Case 3 the heat requirement for solvent regeneration is 2.6 GJ/t $CO_2$, and this demand is related to a technology that is expected to be feasible by 2020 [7]. Current technologies are yet further from this parameter (3.2–3.6 GJ/t $CO_2$) [4,24]. With lower steam requirement per tonne of $CO_2$ captured (1180 kg), capture from combustion would be higher, but the maximum cannot yet be reached: 73% during harvest season. As consequence, total annual capture would be 1.09 Mt $CO_2$ (capture rate of 79%). The impacts on power would correspond to 68% of the total generation during the harvest season and to 37% during the off-season. Due to the higher power consumption during harvest, final surplus electricity decreases to 59 kWh/t of cane.

Finally, in Case 4, with heat requirement equivalent to 726 kg of steam per tonne of $CO_2$, it would be possible to supply all steam required both for the industrial process and the CCS unit, resulting in maximum capture efficiency. In this case the annual capture would be 1.26 Mt $CO_2$, corresponding to a capture rate of 91% (due to fermentation). The net production of energy during the harvest was further reduced (73% of the power output), but the impact during off-season was reduced to 35% of power output. The indicator of surplus electricity per tonne of sugarcane was estimated at 57 kWh.

## 3.2. Economic Performance

In order to compare the results presented in this paper with some presented in the literature, the economic assessment was done with all costs estimated for 2014, in Euro. In all cases the investments were supposed to correspond to a single flow in year 0 of the cash flow.

In the reference case, in which the aim is to maximize surplus electricity, the MSP was estimated taking into account all taxes and charges usually incident to this type of enterprise in Brazil. Table 4 presents the main costs and the calculated MSP for the reference case.

The MSP resulted at 48 €/MWh, a value that could be compared with the reference price set in auctions for new enterprises in 2014 (New Energy Auctions), 62 €/MWh, for biomass power units [25]. The difference is explained by the discount rate usually assumed by investors in bioelectricity (higher than the one assumed here) and by the competition in the electricity sector, which vary depending on the investments in other energy sources and the expectation of investing in new hydro power plants.

Results for the Cases 1–4 are also presented in Table 4. The estimated $CO_2$ credit price, based on the amount of $CO_2$ captured, would cover all costs (the minimum rate of attractiveness would be 10% per year) and also the loss of revenue due to less electricity sold. In this sense, the results presented in Table 4 correspond to the minimum selling price of capturing $CO_2$.

In Case 1, a small amount of $CO_2$ is captured with no meaningful energy penalty. The $CO_2$ minimum credit price was estimated at 21 €/t $CO_2$, a relative low cost for CCS, being the best opportunity in case of sugarcane mills. The estimated $CO_2$ price for Case 1 is basically the same presented in [9] (27.2 US\$/t $CO_2$ in 2014, when the average exchange rate was 1.33 Euro/US\$).

**Table 4.** Costs and economic performance indicators for the BECCS systems.

| Parameter | Reference Case | Case 1 | Case 2 | Case 3 | Case 4 |
|---|---|---|---|---|---|
| Total plant costs | | | | | |
| Power plant (M€) | 77 | 77 | 77 | 77 | 77 |
| $CO_2$ capture unit (M€) | - | - | 171.6 | 224.5 | 253.8 |
| $CO_2$ compression unit (M€) | - | 11.1 | 26 | 33.6 | 37 |
| $CO_2$ transport and storage (M€) | - | 1.3 | 3.0 | 4.0 | 4.5 |
| Fuel costs (M€/y) | 11.4 | 11.4 | 11.4 | 11.4 | 11.4 |
| O&M costs | | | | | |
| Power plant (M€/y) | 1.5 | 1.5 | 1.5 | 1.5 | 1.5 |
| $CO_2$ capture unit (M€/y) | - | - | 10.0 | 13.1 | 14.8 |
| $CO_2$ compression unit (M€/y) | - | 0.5 | 1.2 | 1.5 | 1.7 |
| $CO_2$ transport and storage (k€/y) | - | 26 | 62 | 80 | 89 |
| Performance indicators | | | | | |
| Electricity price (MSP) (€/MWh) | 48 | 48 | 48 | 48 | 48 |
| $CO_2$ credit (minimum price) (€/t $CO_2$) | - | 21 | 66 | 62 | 59 |

It can be seen from Table 4 that, in Cases 2 to 4, the capital costs due to the CCS units represent from 72% to 79% of the total investment, and from 88% to 92% of the total annual O&M costs. It is clear from these figures that carbon capture would be the main driver of investments in Cases 2 to 4, far exceeding the costs of surplus electricity production.

A comparison with the results presented by [4,18] is shown in Table 5. References consider CCS plants based on MEA technology, and both publications were used as reference for estimating costs and performance parameters. However, in both cases the estimates were done for CCS natural gas combined cycle (NGCC) power plants. MEA technology considered in [4] had a heat specific requirement equal to 3.66 GJ/t $CO_2$, that would be intermediate between Cases 2 and 3 in this paper, while [18] considers heat requirement similar to Case 4. As in both references the discount rate is relatively low (7–7.5%), new results related with this study—for a discount rate of 8% per year—were included in Table 5. The minimum price to be paid per tonne of $CO_2$ captured is relatively similar comparing the results of this study (for lower discount rate) to those presented by [18], but the cases are very different for a straight comparison.

**Table 5.** Comparison among similar cases of CCS in thermal power plants.

| Parameter | This Study | | [4] | [18] |
|---|---|---|---|---|
| Electricity production technology | CEST | | NGCC | NGCC |
| Power plant capacity (MW) | 100 | | 830 | 557 |
| Specific heat requirement for MEA (GJ/t $CO_2$) | 2.6 | | 3.66 | 4.4 |
| Total $CO_2$ captured (Mt $CO_2$/y) | 1.09 | | 1.9 | 1.31 |
| Discount rate | 10% | 8% | 7.5% | 7% |
| Base year (for costs) | 2014 | | 2014 | 2011 |
| MSP of electricity (€/MWh) | 48.0 | 44.0 | 90.7 | 49.4 [a] |
| $CO_2$ credit (€/t $CO_2$) | 62.0 | 55.0 | 80.7 | 51.1 [a] |

[a] Original values in US dollar were converted to euro using the average exchange rate in 2011 (0.719).

The more expensive electricity generation is, the higher the credit for capturing $CO_2$ would be. The results of this study are relative close to those presented by [18] because here the case is related to a cogeneration unit that mostly uses residual biomass as fuel, despite the fact that the benefits of scaling effects on electricity generation do not exist. The results presented by [5] are impacted by a higher cost of fuel. Another important aspect is that for the case reported in this paper, the minimum price to be paid for capturing $CO_2$ is impacted by the stream of $CO_2$ from fermentation (that varies from 16% of the total in Case 2, to 10% in Case 4) and that has a relatively small cost. Following the same procedure

described in this paper, but not considering the capture of $CO_2$ from fermentation, the credit price would grow from 59 €/t $CO_2$ to 70 €/t $CO_2$ (for discount rate 10%). It is also worth mentioning that the amount of $CO_2$ captured per year is not much smaller (57% to 83%) than in power units that would burn natural gas, despite the much smaller installed electricity capacity (12% to 18%). Comparing biomass and natural gas, the higher carbon content per unit of energy and the much lower efficiency of electricity generation explain the huge penalties of $CO_2$ capture on electricity generation. Another useful comparison is with the carbon price needed for making a technology competitive. In the case of switching from NGCC to a coal power plant with CCS, considering 8% as the annual capital costs for the investments, [26] presents 85 €/t $CO_2$ as the break-even price. Thus, in a general sense it can be concluded that the $CO_2$ capture costs presented in this paper are in line with the estimates presented in the literature, but a main difference is that the BECCS system considered here is able to contribute with negative GHG emissions.

Cases 2 to 4 correspond to different stages of development of MEA based technology. Case 2 corresponds to the current commercial stage, while Case 3 represents the technology that could be available in short term. Moving from current to future technologies would impact carbon capture, with an increase of 38% on annual output as long as Case 3 is compared to Case 2; as previously mentioned, capturing in Case 2 is negatively impacted by the higher steam demand for recovering amine. On the other hand, moving from Case 2 to Case 3 significantly impact surplus electricity, with a decrease of almost 13% in the total electricity that could be sold along the year. The impact on the minimum price to be paid per tonne of $CO_2$ captured is less pronounced, with a reduction of 6% comparing Cases 3 and 2. The change from Case 3 to Case 4 is less pronounced (an increase of 16% in annual capture, a reduction of 2.8% in surplus electricity, and a 4.8% reduction in carbon costs).

The fact that the minimum price to be paid per tonne of $CO_2$ captured is almost equal in all three cases indicates that, from an economic point of view, it is not necessary to wait for advanced MEA technologies in order to go for pilot BECCS projects. Therefore, it is necessary to consider technologies that would impact less on surplus electricity. However, a very important find is that $CO_2$ capture from fermentation has a lower cost and a small impact on the energy balance, and should be prioritized for pilot BECCS units in Brazil.

*3.3. Scaling Effects*

For the considered BECCS system scaling effects are analyzed in this section. Case 3, with heat requirement for solvent regeneration equivalent to 2.6 GJ/t $CO_2$, was chosen to be scaled into a smaller industry (2 Mt/y) and a larger mill (8 Mt/y). The same performance parameters previously presented were considered, and costs were estimated according to the assumptions mentioned before (for 10% discount rate). Table 6 presents total plant costs and the main economic results. Scale effects are clear both on the MSP of surplus electricity and on the minimum price to be paid for capturing $CO_2$.

Taking as reference the price presented by [4] in the case of capture in a large combined cycle power plant (80.7 €/t $CO_2$), and assuming that people would be able to pay this value in the future, full carbon capture (both from fermentation and combustion) would be feasible in Brazilian sugarcane mills, but it is clear that the feasibility would be enhanced with the mill capacity. As regard with the results presented by [18], comparatively the feasibility would exist for larger mills. Considering that mills with capacity equivalent to 2 Mt/y are currently the average in Brazil, in many existing mills it would be feasible to capture $CO_2$. On the other hand, considering that new mills tend to be larger, in the future it would be reasonable to consider mill's location also taking into account the aim of storing $CO_2$ at lower costs.

Allocating all capital costs to the annual surplus electricity, the indicator varies from 1874 to 1099 €/MWh, depending on the mill size (see Table 6). This figure for the reference case [4] is only 134 €/MWh. Alternatively, allocating total capital costs to the annual amount of $CO_2$ captured, this indicator varies from 237 to 405 €/t $CO_2$ captured per year (Table 6), while this figure is 483 €/t in the case presented by [4]. It seems clear that investors should have a completely different

rationale in each case: while in case of a natural gas combined power plant $CO_2$ capture would be a complement that should be fairly paid, in the case of carbon capture in a sugarcane mill surplus electricity produced in a cogeneration unit should no longer be the priority. In this case, the priority should be capturing $CO_2$, with the advantage that this enterprise would contribute with negative emissions. Indeed, whether the benefits of negative emissions would be recognized, a larger payment per tonne of $CO_2$ would be possible, and surplus electricity could be more competitive with other generation options.

**Table 6.** Total investment costs and economic performance results for different milling capacities.

| Parameter | Milling Capacity (Mt/y) | | |
|---|---|---|---|
| | 2 | 4 | 8 |
| Performance results | | | |
| Power plant capacity (MW) | 50 | 100 | 200 |
| $CO_2$ captured (Mt $CO_2$/y) | 0.55 | 1.09 | 2.19 |
| Electricity price (MSP) (€/MWh) | 54 | 48 | 43 |
| $CO_2$ credit (minimum price) (€/t $CO_2$) | 80 | 62 | 48 |
| Total plant costs | | | |
| Power plant (M€) | 48.5 | 77.0 | 122.3 |
| Capture unit (M€) | 148.1 | 224.5 | 340.3 |
| Compression unit (M€) | 22.2 | 33.6 | 50.9 |
| Transport and storage (M€) | 2.7 | 4.0 | 6.1 |
| Economic indicators (as function of annual outputs) | | | |
| Investment cost per tonne of $CO_2$ captured (€/t) | 405 | 310 | 237 |
| Investment cost per surplus electricity unit (€/MWh) | 1874 | 1435 | 1099 |

## 4. Conclusions

This work aimed to explore the technical and economic feasibility of BECCS systems in the Brazilian sugarcane sector. Post-combustion technology based on MEA was considered for capturing $CO_2$ from biomass combustion, and three technology levels—related to heat requirements for solvent regeneration—were assessed. Results show that $CO_2$ capture, both from fermentation and combustion, is technically possible but energy penalties are meaningful in case of combustion, with considerable impacts on surplus electricity. Energy penalties due CCS imply deep reduction in electricity generation, varying from 43% to 52% regarding the reference case. In this sense, it is important to evaluate other technologies for capturing $CO_2$. The more expensive the electricity sold, the higher the price to be paid per tonne of $CO_2$ captured.

Comparatively, capturing $CO_2$ from the fermentation is the best opportunity, because of the low impact on the mill, the relatively low cost, and the benefits on the ethanol carbon footprint. Clearly, is the alternative that should be prioritized.

The high impact on electricity production in the case of biomass-based cogeneration units, compared with well-known estimates for natural gas plants, is due to the comparatively low efficiency of electricity generation. Investments and costs are far greater for capturing $CO_2$ than for generating surplus electricity. In this sense, the rationale for investments should be different: the priority would be capturing $CO_2$—resulting in net negative emissions, and selling electricity would be a second priority.

The $CO_2$ credits presented in this document and also in the literature (approximately € 45–80/t $CO_2$) are much higher than the current price of carbon in different markets (e.g., considering the $CO_2$ European Emission Allowances, the carbon price was less than € 10/t $CO_2$ in the first half of 2018 and around € 20/t $CO_2$ by the end of 2018 [27]), but it is important to take into account some important aspects. First, carbon markets are currently depressed due to low demand. Second, and most importantly, carbon capture and storage would be costly among the mitigation options and the large-scale implementation of CCS systems would be feasible only in the medium to long term.

Supposing that carbon capture through CCS would be a target in the future, investing in a BECCS system in a sugarcane mill would be much more effective than investing in a power plant that burns natural gas, for instance. The price to be paid would be lower, and the result would be negative emissions.

**Author Contributions:** Conceptualization, S.R.V. and A.W; Methodology, S.R.V. and A.W; Software, S.R.V; Validation, S.R.V. and A.W; Investigation, S.R.V. and A.W; Writing-Original S.R.V; Draft Preparation, S.R.V; Writing-Review & Editing, S.R.V. and A.W; Visualization, S.R.V; Supervision, A.W.

**Funding:** This research received no external funding.

**Acknowledgments:** The authors would like to acknowledge the support of CAPES (Coordenação de Aperfeiçoamento de Pessoal de Nível Superior, Brazil).

**Conflicts of Interest:** The authors declare no conflict of interest.

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
