# Peer review of "Techno-Economic Assessment of Bio-Energy with Carbon Capture and Storage Systems in a Typical Sugarcane Mill in Brazil"

_energies, doi:10.3390/en12061129_

Round 1

Reviewer 1 Report

The article present a case study of BECCS potential in the Brazilian sugarcane industry. The topic is novel and interesting, although the work presents some shortcoming and requires consistent revision. 

1) The analysis have been made only at plant level, with missing/simplistic assumptions on important part of a BECCS "system"/supply chain (e.g. biomass sources, emission released during the transport of biomass, safety of the selected storage site...etc). In this sense, the title of the work is misleading. 

2) The boundaries of the system investigated are not clear: from an environmental point of view it seems that the authors are investigating only the technology aspects of the BECCS systems (emission released during the combustion process) while the cost of biomass transport are included in the economic assessment... Maybe, a graphical scheme describing the boundaries of the system investigated would help. 

3) Methods: What is the real contribution of the work? the methodology is poorly described. In the present form, it seems that the authors have performed some feasibility assessment of BECCS potential by collecting some available literature data. What software has been used for the analysis? This is barely mentioned and should be the focus of this session.  

4) The biggest shortcoming is that the paper relies on the assumption that the biomass is carbon neutral. This is a very strong assumption that need to be justified (where is the biomass coming from? what assumptions have been made with regard to the CO2 emissions released during the transport of biomass?). This is  very important also in light of the sustainability issues connected with BECCS which is receiving growing criticism with respect of its ability to deliver negative emissions. 

5) Results analysis is missing. Given that the input data have been mainly taken from literature, a sensitivity analysis of the results would be needed. 

Author Response

Response to Reviewer 1 Comments*

*Comments received in black and answers in blue

Reviewer # 1

The article present a case study of BECCS potential in the Brazilian sugarcane industry. The topic is novel and interesting, although the work presents some shortcoming and requires consistent revision

The manuscript "Techno-economic evaluation of BECCS systems in the Brazilian sugarcane sector" was presented at the 13th Conference on Sustainable Development of Energy, Water and Environment Systems - SDEWES Conference, held in Palermo, in 2018. The authors were invited to present a paper to Energies, being this submission closely related to the text presented at the 13th SDEWES. Respecting the instructions, the authors chose to make improvements, but also maintaining the essentials of what has been previously presented. In addition to following the instructions, there was a limited time for both the initial submission and the one after the first round of revisions.

Thanks for the comments and suggestions. The authors have worked on incorporating all suggestions to improve the manuscript. Below are the answers for each of the comments received.

·         Comment # 1

The analysis has been made only at plant level, with missing/simplistic assumptions on important part of a BECCS "system"/supply chain (e.g. biomass sources, emission released during the transport of biomass, safety of the selected storage site...etc). In this sense, the title of the work is misleading.

Response 1: The evaluation was carried out for a typical sugar cane mill, which size is representative of the bulk of existing mills in Brazil. Average parameters have been considered throughout the analysis. In addition to that, knowing that the scale has a crucial impact on viability, the results of a simple sensitivity analysis are presented, considering smaller and larger plants. At the beginning of section 2, Materials and methods, a paragraph was included to clarify in advance the simplifications made and how representative the results would be. The authors believe that the results are representative of the Brazilian sugarcane sector but, to avoid any misinterpretation, the title was changed to "Techno-economic evaluation of the BECCS systems in a typical sugarcane mill in Brazil".

·         Comment # 2

The boundaries of the system investigated are not clear: from an environmental point of view it seems that the authors are investigating only the technology aspects of the BECCS systems (emission released during the combustion process) while the cost of biomass transport are included in the economic assessment... Maybe, a graphical scheme describing the boundaries of the system investigated would help.

Response 2: Yes, the reviewer has understood correctly. The authors have tried to create a graphic scheme to describe the limits but, being honest, they could not conceive anything different from a very simple and obvious graphic, and, in this sense, unnecessary.

·         Comment # 3

What is the real contribution of the work? The methodology is poorly described. In the present form, it seems that the authors have performed some feasibility assessment of BECCS potential by collecting some available literature data. What software has been used for the analysis? This is barely mentioned and should be the focus of this session.

Response 3: As far as the authors know, there is no published paper with similar results. In addition to this contribution, the conclusion that it would be possible to capture CO2 at a cost significantly lower than the cost of capture in conventional thermal plants is, from the point of view of the authors, an important result. Yes, the paper is based on feasibility assessment, based on available high-quality literature.

The methodology section has been improved, partly due to the actions in response to the other two reviewers (for example, additional information in 2.1, 2.3 and 2.3.3), and with a new section (2.5) in order to clarify the simplified hypothesis related to GHG emissions due to the supply of biomass (bagasse and straw).

The authors have used a non-commercial software for evaluating electricity generation, and this is mentioned in 2.1, line 101. The authors have developed their own procedure for calculating the interface with the CCS plant, the energy penalties, and for developing the economic assessment. The authors that this information is not necessary along the paper.

The authors are thankful for the comment related to the methodology.

·         Comment # 4

The biggest shortcoming is that the paper relies on the assumption that the biomass is carbon neutral. This is a very strong assumption that need to be justified (where is the biomass coming from? what assumptions have been made with regard to the CO2 emissions released during the transport of biomass?). This is  very important also in light of the sustainability issues connected with BECCS which is receiving growing criticism with respect of its ability to deliver negative emissions.

Response 4: Explanation about this assumption was included in a new section, please see section 2.5. As simplification, there was assumed that both bagasse and straw are carbon neutral.

·         Comment # 5

Results analysis is missing. Given that the input data have been mainly taken from literature, a sensitivity analysis of the results would be needed.

Response 5: The results and the discussion of the results are presented throughout section 3. As mentioned, capital costs and O&M costs, in addition to the required performance parameters, were taken from high-quality papers that present similar results – but in different thermal power plants – to the case considered in this document. And the results presented in this document were compared with those presented in the literature (e.g. see Table 5 and the discussion related to the results presented). Obviously, the comparison is only possible when the technology is the same, and provided that the costs correspond to the same year. In this sense, it would not make sense to compare the results without respecting the most basic hypotheses. Considering the whole set of results presented, there are three simple sensitivity analysis, but with discrete results: (1) considering the range of energy requirements of the capture technology; (2) considering the size of the sugar cane mill; (3) considering different discount rates. Therefore, no other action was taken.

Reviewer 2 Report

BECCS applied to sugar cane mills producing ethanol and co-generation from biomass combustion. Cost and feasibility of state of the art commercially available CC technologies are evaluated and decarbonising fermentation producing a pure CO2 stream (compression only) is less energy intensive than decarbonising combustion diluted CO2 gas streams ( absorption, regeneration and compression). Any estimated solution as an unavoidable cost attached to it, the cheapest 48 euro/tonne for large scale mills. High energy penalty of CO2 capture is also calculated to be ab8ut 50% depending on the phase of operation. Capture clearly emerges as the most expensive step for all mill sizes evaluated in the study. Storage is the cheapest instead, although little is examined in that terms.

The paper is well written and structured. Timely and relevant, it can be published once the following aspects are addressed in full:

Page 2-Line 81: Add a brief but informative comment on the size, footprint, environmental impact and cost of storage of biomass for the off-season power generation,

Page 3-Line 100: Add a brief explanation of why straw different from bagasse causes slagging, fouling and surface corrosion.

Page 4-Line 149: Provide more information about the minimum credits to be earned by selling the capture service. What is it meant in practice about selling the capture service?

Section 2.3.3: Separated CO2 compressed to 150 bar transported to a saline aquifer for storage, what is the expected length of long term storage? What risk does this introduce in the real implementation of carbon negative BECCS? This is essentials to the overall value of the study, and as now it is the major weakness of the manuscript.

Author Response

Response to Reviewer 2 Comments*

*Comments received in black and answers in blue

Reviewer # 2

BECCS applied to sugar cane mills producing ethanol and co-generation from biomass combustion. Cost and feasibility of state of the art commercially available CC technologies are evaluated and decarbonising fermentation producing a pure CO2 stream (compression only) is less energy intensive than decarbonising combustion diluted CO2 gas streams (absorption, regeneration and compression). Any estimated solution as an unavoidable cost attached to it, the cheapest 48 euro/tonne for large scale mills. High energy penalty of CO2 capture is also calculated to be ab8ut 50% depending on the phase of operation. Capture clearly emerges as the most expensive step for all mill sizes evaluated in the study. Storage is the cheapest instead, although little is examined in that terms.

The paper is well written and structured. Timely and relevant, it can be published once the following aspects are addressed in full:

The manuscript "Techno-economic evaluation of BECCS systems in the Brazilian sugarcane sector" was presented at the 13th Conference on Sustainable Development of Energy, Water and Environment Systems - SDEWES Conference, held in Palermo, in 2018. The authors were invited to present a paper to Energies, being this submission closely related to the text presented at the 13th SDEWES. Respecting the instructions, the authors chose to make improvements, but also maintaining the essentials of what has been previously presented. In addition to following the instructions, there was a limited time for both the initial submission and the one after the first round of revisions.

Thanks for the comments and suggestions. The authors have worked on incorporating all suggestions to improve the manuscript. Below are the answers for each of the comments received.

·         Comment # 1

Page 2-Line 81: Add a brief but informative comment on the size, footprint, environmental impact and cost of storage of biomass for the off-season power generation.

Response 1:  A comment was added in page 2, lines 83-84. In summary, storage of biomass for the off-season is a common practice in some mills.

·         Comment # 2

Page 3-Line 100: Add a brief explanation of why straw different from bagasse causes slagging, fouling and surface corrosion.

Response 2:  The reported problems are due to straw and bagasse compositions, and also due their ash compositions: straw has much more chlorine, CaO and K2O than bagasse. An explanation was included in lines 104 to 106.

·         Comment # 3

Page 4-Line 149: Provide more information about the minimum credits to be earned by selling the capture service. What is it meant in practice about selling the capture service?

Response 2:  There was an error in the previous version and "minimum credit price" should be written in place of "minimum credits". The sentence was rewritten (please see lines 159-161).

·         Comment # 4

Section 2.3.3: Separated CO2 compressed to 150 bar transported to a saline aquifer for storage, what is the expected length of long term storage? What risk does this introduce in the real implementation of carbon negative BECCS? This is essentials to the overall value of the study, and as now it is the major weakness of the manuscript.

Response 2:  The rationale of the CCS as a mitigation option is that carbon dioxide would be stored for centuries, in this case, in a saline aquifer. The existence of adequate geological conditions is the main issue and, in the case of the saline aquifer below the Guaraní reservoir, it is known that suitability exists. Obvious that there are uncertainties, and the risk of leakage. A comment was included on lines 185 to 186.

Reviewer 3 Report

The paper is well rounded analysis of CO2 mitigation from a typical industrial element (sugarcane mill) that considered cogeneration technologies, economics and CO2 storage.

I only recommend few improvements:

- nomenclature in equations

- I'm not sure about some terms (related to penalties), eg. compression penalty. The terms hould be revised or at least explained (why there is penalty for commpression?!?)

- some numbers seem to be outdated. The most critics I'd put to CO2 credit, as the source is not quite clear, and 62 €/metric ton of CO2 (or 80.7 in case 5, table 5) seems to change the whole analysis. Please consider the CO2 price, introduce more CO2 price scenarios or provide adequate explanation.

- discount rate of 10% should be (at least) discussed. I think the results are not hard to modify, according to several discount rates. 10% for intensive industries (and CCS is related to such industries) seems to be rather optimistic.

Author Response

Response to Reviewer 3 Comments*

*Comments received in black and answers in blue

Reviewer # 3

The paper is well rounded analysis of CO2 mitigation from a typical industrial element (sugarcane mill) that considered cogeneration technologies, economics and CO2 storage.

I only recommend few improvements:

The manuscript "Techno-economic evaluation of BECCS systems in the Brazilian sugarcane sector" was presented at the 13th Conference on Sustainable Development of Energy, Water and Environment Systems - SDEWES Conference, held in Palermo, in 2018. The authors were invited to present a paper to Energies, being this submission closely related to the text presented at the 13th SDEWES. Respecting the instructions, the authors chose to make improvements, but also maintaining the essentials of what has been previously presented. In addition to following the instructions, there was a limited time for both the initial submission and the one after the first round of revisions.

Thanks for the comments and suggestions. The authors have worked on incorporating all suggestions to improve the manuscript. Below are the answers for each of the comments received.

·         Comment # 1

nomenclature in equations

Response 1: Nomenclature was included after equations (1) and (2).

·         Comment # 2

I'm not sure about some terms (related to penalties), e.g. compression penalty. The terms should be revised or at least explained (why there is penalty for compression?!?)

Response 2: In the literature, “penalty” is used to express energy and power requirements related to a CCS plant. The energy demand (thermal and electricity) imposes penalties to the system. In page 2, line 47, an explanation was included. In Table 3 the expression “compression penalty” was changed to “compression power”.

·         Comment # 3

Some numbers seem to be outdated. The most critics I'd put to CO2 credit, as the source is not quite clear, and 62 €/metric ton of CO2 (or 80.7 in case 5, table 5) seems to change the whole analysis. Please consider the CO2 price, introduce more CO2 price scenarios or provide adequate explanation.

Response 2: The value 62 €/t of CO2 is a result of the submitted paper in the reference case. The result is the minimum selling price for covering the costs along the life cycle. On the other hand, 51.1 and 80.7 €/t of CO2 are similar results presented in the literature, for a similar CCS technology (MEA), but to be used in natural gas combined cycles. They are estimated minimum selling price for the best available information. Table 5 presents these values, and the main parameters of each case; the intention is the comparison, first to validate the results, and second, to support the analysis that is presented along this part of the text. For the time being there is no market for selling credits of carbon capture and any scenario would be speculative. Nowadays, the price paid per unit of avoided CO2 – regardless the technology considered – is very low compared to the results presented, due to the current status of actions for mitigating GHG emissions. In the submitted paper, the argument is that whenever these markets exist, the minimum price of 62 €/t of CO2 would be attractive, first because of the lower value, and second because capture would be related to negative GHG emissions, and not just mitigation. Additional information about carbon prices was included in “Conclusions”, lines 400 to 406.

·         Comment # 4

Discount rate of 10% should be (at least) discussed. I think the results are not hard to modify, according to several discount rates. 10% for intensive industries (and CCS is related to such industries) seems to be rather optimistic.

Response 3: Discount rate of 10% is the reference case in this paper (results are presented in Tables 3 and 4) but results for 8% are also presented in Table 5. The reason for considering 10% in the base case is because the authors knew in advance the impact of not-selling electricity in the final results. In Brazil, investors for generating electricity from sugarcane residues would not accept discount rate lower than 10%. On the other hand, investments on CCS would require – at least for the first plants – discount rate lower than 10% (e.g. 7-8%). As most of the results available in the literature are for discount rates close to 8%, and the intention was the comparison, the authors decided to explore the sensitivity analysis only for this rate. Information was added in section 2.3, lines 150-155.

Round 2

Reviewer 1 Report

Previous concerns have been addressed. In my view, the article can be accepted in present form. 

Reviewer 2 Report

The authors have addressed all concerns. The manuscript is now clearer and from my side it can be published.

Reviewer 3 Report

The authors gave satisfactory responses to my comments, the only critical thing is analysis for different discount rates, but if other reviewers and editors haven't found this part critical, I'd recommend the paper for publishing.